# Obesity-Induced Brain Neuroinflammatory and Mitochondrial Changes

**DOI:** 10.3390/metabo13010086

**Published:** 2023-01-05

**Authors:** Luisa O. Schmitt, Joana M. Gaspar

**Affiliations:** 1Laboratory of Neuroimmuno-Metabolism, Federal University of Santa Catarina, Florianopolis 88040-900, SC, Brazil; 2Graduate Program in Biochemistry, Federal University of Santa Catarina, Florianopolis 88040-900, SC, Brazil

**Keywords:** high-fat diets, cognitive decline, metabolic disorders, energy homeostasis, neuroinflammation, mitochondria, hippocampus, hypothalamus

## Abstract

Obesity is defined as abnormal and excessive fat accumulation, and it is a risk factor for developing metabolic and neurodegenerative diseases and cognitive deficits. Obesity is caused by an imbalance in energy homeostasis resulting from increased caloric intake associated with a sedentary lifestyle. However, the entire physiopathology linking obesity with neurodegeneration and cognitive decline has not yet been elucidated. During the progression of obesity, adipose tissue undergoes immune, metabolic, and functional changes that induce chronic low-grade inflammation. It has been proposed that inflammatory processes may participate in both the peripheral disorders and brain disorders associated with obesity, including the development of cognitive deficits. In addition, mitochondrial dysfunction is related to inflammation and oxidative stress, causing cellular oxidative damage. Preclinical and clinical studies of obesity and metabolic disorders have demonstrated mitochondrial brain dysfunction. Since neuronal cells have a high energy demand and mitochondria play an important role in maintaining a constant energy supply, impairments in mitochondrial activity lead to neuronal damage and dysfunction and, consequently, to neurotoxicity. In this review, we highlight the effect of obesity and high-fat diet consumption on brain neuroinflammation and mitochondrial changes as a link between metabolic dysfunction and cognitive decline.

## 1. Introduction

Obesity is a chronic disease defined as abnormal and excessive fat accumulation, and it represents an important risk factor for many diseases and premature death. Body mass index (BMI) is a weight-for-height index commonly used to classify overweight and obesity in adults. The healthy weight range is a BMI range between 18.5 and <24.9 kg/m^2^. If an individual’s BMI is between 25.0 and 29.9 kg/m^2^, they fall within the overweight range; if their BMI is 30.0 kg/m^2^ or higher, they are considered obese. In 2015, more than 1.9 billion adults were overweight; over 600 million were obese [1,2]. Even more alarming is that child obesity affects 107.7 million children [1,2]. Changes in lifestyle in the last century (increased consumption of hypercaloric diets and sedentary behavior) are the fundamental causes of obesity epidemics.

Obesity is associated with an increase in noncommunicable diseases, including metabolic and cardiovascular diseases, some types of cancer, musculoskeletal disease, and several brain diseases, which represent the leading causes of premature mortality and disability [2,3,4]. In addition, long-term high-fat diet (HFD) consumption has been found to induce peripheral insulin resistance and cause brain insulin resistance [5,6,7].

In recent years, increasing attention has been given to the relationship of obesity and associated insulin resistance/type 2 diabetes with the development of brain diseases, including depression, neurodegenerative diseases, dementia, and vascular dementia [8,9,10]. Epidemiological studies have shown that people with a higher BMI are at greater risk for developing Alzheimer’s disease than subjects with normal BMI [11,12,13]. Moreover, some population-based studies have identified diabetes as a risk factor for dementia and metabolic syndrome, a grouping of risk factors for type 2 diabetes mellitus [14,15,16]. However, cellular and molecular mechanisms linking these conditions have not yet been fully elucidated.

Since neuronal cells have a high energy demand, the mitochondrial machinery plays an important role that ensures a constant energy supply in order to guarantee the function of these cells. In this review, we focused on the effect of obesity on brain neuroinflammation and mitochondrial changes as a link between obesity and cognitive impairments.

## 2. Mitochondria Functions and Dynamics

Mitochondria are double-membrane organelles responsible for energy production and homeostasis, the regulation of intracellular calcium levels, and the regulation of apoptosis (mainly via the intrinsic pathway) [17,18]. In addition, mitochondria are responsible for generating more than 90% of the energy for the cell through oxidative phosphorylation [19,20].

To generate ATP through oxidative phosphorylation, mitochondria use an electron transport chain inserted within the mitochondrion’s inner membrane (Figure 1) [21]. NADH and FADH_2_ are generated by the Krebs cycle and donate electrons to complex I (NADH: ubiquinone oxidoreductase) and complex II (succinate dehydrogenase), respectively. The electrons from NADH are passed from complex I to ubiquinone (CoQ) in order to enter the Q cycle, where CoQ is reduced to ubiquinol (QH_2_). This electron transfer induces the pumping of protons by complex I from the matrix into the intermembrane space. The electrons donated from FADH_2_ are transferred from complex II to CoQ similarly to complex I, although this process is not accompanied by proton translocation [21]. Once in the Q cycle, the electrons are transferred to complex III (coenzyme Q: cytochrome c reductase) and then to cytochrome c, releasing two protons into the intermembrane space. Then, when cytochrome c is reduced, it transports single electrons from complex III to complex IV (cytochrome c oxidase), where molecular oxygen is reduced to water. At complex IV, a total of eight protons are pumped from the matrix, of which four are used to form two water molecules, and the other four are transferred into the intermembrane space [21,22].

In response to electron transport, a total of ten protons are pumped from the matrix into the intermembrane space, where they accumulate to generate an electrochemical and concentration proton gradient that generates a proton motive force, essential for the activity of complex V (ATP synthase) to generate ATP [21]. A consequence of electron transfer is the generation of reactive oxygen species (ROS), which contributes to homeostatic signaling. However, when ROS are produced in excess, they cause oxidative stress and can lead to mitochondrial dysfunction and diseases [22]. Therefore, an efficient measurement of the electron transport chain function and ATP production, using high-resolution respirometry, such as a Seahorse XF24 Extracellular Flux Analyzer and oxygraphy, can provide insight into cellular physiology and dysfunction.

Mitochondria are highly dynamic organelles that undergo a continuous cycle of fission and fusion, processes called mitochondrial dynamics (Figure 1). Another dynamic process of mitochondria is the selective removal of dysfunctional mitochondria, a quality-control mechanism that ensures a healthy mitochondrial population. The dynamic properties of mitochondria are critical for their optimal function in energy generation [23]. Mitophagy is a mechanism of mitochondrial quality control used to eliminate damaged mitochondria and prevent excessive ROS production, thus maintaining homeostasis in mitochondria. Mitochondrial dynamics involve the plasma membrane and organelles, such as ER and lysosomes. The contact point of ER–mitochondria is referred to as mitochondria-associated ER membranes. Some studies have suggested that the integrity of mitochondria-associated ER membranes is required for insulin signaling (for a detailed description, see the revision [24]). Studies have been carried out to investigate the effect/defect of insulin signaling on different features of mitochondrial dysfunction, focusing on dynamics, biogenesis, and mitophagy and their role in pathologies in which metabolic dysmetabolism is comorbid with neurodegeneration [25,26]. Some studies have also suggested that the protective actions of leptin may be facilitated through the regulation of mitochondrial dynamics, namely, mitochondrial fission and fusion [27,28,29,30].

Dysfunctional mitochondria are recognized by the autophagy machinery, resulting in their engulfment by autophagosomes and trafficking to the lysosome for degradation. The most common mitophagy pathways are mediated by PINK1 and PARKIN proteins. Mitochondrial fission is the process where mitochondria divide into two separate mitochondrial organelles. Fission is mediated by the interaction between the mitochondrial fission factor (Mff) and dynamin-related protein-1 (Drp1). Briefly, Drp1 is recruited from a cytosolic pool onto the mitochondrial surface, where it self-assembles into spiral structures to facilitate fission, acting similarly to endocytic invaginations of the cell membrane. Several mitochondrial-bound proteins then aid in the recruitment of Drp1 to the mitochondria, including Fis1, Mff, MiD49, and MiD51 [23,31,32].

Fusion is the process of joining two adjacent mitochondria through a physical merging of the outer and then the inner mitochondrial membranes, resulting in the content mixing of the matrix components diffusing throughout the new mitochondrion. Fusion is mediated by the proteins mitofusin-1 (Mfn1) and mitofusin-2 (Mfn2) [33,34,35], located on the mitochondrial outer membrane. Mitofusins are required for outer membrane fusion. The fusion of the inner membrane is mediated by the protein optic atrophy 1 (Opa1), which is associated with the inner membrane ([36] for a detailed description of the mitochondria dynamics, please read the review manuscript [23]).

Mitochondrial dynamics is important for growth redistribution and maintenance in a healthy mitochondria network and plays a role in disease-related processes. All the cells consume energy for their homeostasis and specific activity, and they require the support of functional mitochondria that provide ATP obtained via oxidative phosphorylation. A reduction in mitochondria respiration and bioenergetics is associated with insulin resistance [24].

Therefore, the dysfunction of mitochondrial dynamics and function could lead to disorders in mitochondria, which are greatly associated with the progression of several diseases, including obesity and metabolic and neurological conditions.

## 3. Obesity Induces Cognitive Decline

Obesity, as well as HFD diet consumption, and metabolic disorders, such as diabetes mellitus, are widely recognized as inducing impairments in brain structure and function in the form of memory dysfunction, as well as neurodegenerative diseases [37]. Furthermore, magnetic resonance imaging studies have demonstrated that regional brain atrophy and changes in gray and white matter are observed in patients with obesity, providing new insights into the relationship between obesity and cognitive decline from the imaging perspective [37,38,39,40]. Furthermore, a higher BMI is correlated with a lower gray matter volume in the prefrontal, temporal, insular, and occipital cortexes; thalamus; putamen; amygdala; and cerebellum, mediating the negative effects on memory performance [41].

Patients with obesity have an earlier onset of Alzheimer’s, which is considered an aging disease [42]. An 18-year follow-up longitudinal study demonstrated a higher degree of overweight in older women who developed AD. No associations were found in men [42]. In the same study, the authors concluded that Alzheimer’s disease risk increased by 36% for every 1.0 increase in BMI. In other studies, it has been shown that patients with a higher BMI present significantly lower scores in cognitive tests and a longitudinal decline in cognitive abilities in both men and women [37,43,44]. Changes in cognitive function can be potentiated since middle-aged adults with obesity may experience differentially greater brain atrophy [37]. The relationship between a higher BMI and reduced cognitive performance does not change with age [45] or race [46]. A high intake of fat and sugar is associated with impairments in hippocampal-dependent learning and memory in children [47] and adults [48,49], suggesting a negative impact on hippocampal function across the lifespan. In the community-based Framingham Offspring Cohort, it was observed that central obesity was significantly related to poorer performance in executive function and visuomotor skills, and no changes were observed for verbal memory [50]. Adults with overweight and obesity also have poorer executive function than normal-weight adults, without changes in performance on attention tests. Children and adolescents with overweight/obesity also present poor cognitive function on verbal, full-scale, and performance IQ; visual–spatial; and executive function tests [51,52]. A systemic review found that executive dysfunction is associated with obesity-related behaviors in children and adolescents, such as increased food intake, disinhibited eating, and less physical activity. In children and adolescents, obesity is associated with poorer cognitive competence and may affect their academic achievements [53].

Body weight and diet composition are modified risk factors for cognitive decline. Weight loss appears to be associated with low-order improvements in executive/attention functioning and memory in individuals with obesity. Moreover, a stable BMI predicts better cognitive trajectories [54]. Patients with severe obesity may obtain immediate verbal and delayed memory function benefits from Roux-en-Y gastric bypass [55,56].

Different animal models of obesity and metabolic disorders have also exhibited cognitive dysfunctions and worse performance in learning and memory tasks compared to non-obese animals [57,58,59,60,61,62]. In addition, based on studies of animal models and in vitro models, high levels of glucose and saturated fatty acids are responsible for neuroinflammation, microglia activation, mitochondrial dysfunction, neuronal loss, and impairments in synaptic plasticity (Figure 2) [63,64,65,66,67,68].

### 3.1. Obesity-Induced Cognitive Decline: Role of Neuroinflammation

Obesity is a low-grade chronic inflammatory disease that increases susceptibility to the numerous conditions associated with it. During the expansion of white adipose tissue, the recruitment and infiltration of immune cells, mainly macrophages, occur [69]. The growth of adipose tissue is also associated with an increased expression of proinflammatory cytokines, particularly interleukin-6 (IL-6), interleukin-1 beta (IL-1β), and tumor necrosis-α (TNF-α) [70,71,72]. Subjects with obesity have high circulating proinflammatory adipocytokines that trigger chronic inflammation. Systemic low-grade chronic inflammation has been reported to cause neuroinflammation and changes in different brain structures, such as the cerebellum, amygdala, cerebral cortex, and hypothalamus [57,73,74,75]. Obesity-induced inflammation has been related to changes in the integrity of blood–brain barrier permeability, inducing leukocyte extravasation, along with the potential entry of pathogens and toxins into the central nervous system, which, in turn, stimulate more inflammatory responses in a vicious cycle [57]. The decrease in tight junction protein expression and the disturbed blood–brain barrier are regulated by the NF-κB pathway, which increases the expression of proinflammatory proteins, such as IL-1β, TNFα, and IL-6. The loss of the blood–brain barrier during obesity facilitates proinflammatory cytokines to enter the brain parenchyma, thus allowing them to interact and activate glial cells (microglia and astrocytes) [76]. Activated microglia secrete more inflammatory cytokines (TNFα, IL-1β, and IL-6), perpetuating the neuroinflammation and leading to neuronal damage. NLRP3 proteins of the inflammasome secreted by visceral adipose tissue directly activate microglia through the IL1 receptor [77].

Furthermore, HFD can directly activate microglial cells, inducing morphological changes in the hypothalamus without causing microglial changes in the cerebral cortex and striatum. Moreover, HFD-induced obesity is associated with an increased entry of peripheral immune cells into the central nervous system and may contribute to the inflammatory response [78]. Fatty acid intake induces the activation of immune cells and the inflammatory response through the activation of the innate immune system through Toll-like receptors (TLRs) [63]. The binding of fatty acids to TLR4 activates nuclear factor-Κb (NF-Κb) and activator protein 1 (AP-1), which, in turn, upregulate the expression of proinflammatory cytokines and chemokines [79]. Another proposed mechanism for obesity-induced inflammation relies on the ability of an HFD to modulate the gut microbiota. Indeed, the subsequent changes in microbiota populations result in the permeabilization of the gut barrier, leading to the increased passage of bacterial endotoxins into the circulation [80,81].

Based on studies conducted on animal models of obesity, it has been proposed that inflammatory processes may participate in both the peripheral and brain disorders associated with obesity, including the development of cognitive alterations. Diet-induced obesity leads to microglia activation, which induces synaptic alterations, including impairment in hippocampal synaptic plasticity, reductions in dendritic spine density and the sites of excitatory synapses, and promoted synaptic stripping [58].

#### 3.1.1. Hypothalamic Neuroinflammation

The hypothalamus is a master regulator of whole-body energy homeostasis and metabolism (see review [82]) through hormonal and nutrient-sensing mechanisms. The integrity of the hypothalamic nuclei that regulate appetite satiety is altered by neuroinflammation. Many studies have shown that exposure to obesity and HFD feeding strongly affects the hypothalamic mechanisms of energy homeostasis regulation [73,83,84,85].

Diet-induced obesity increases the activation of inflammation in the mediobasal hypothalamus, resulting in the production of proinflammatory cytokines (TNF-α, IL-1β, and IL-6) and impairment in insulin and leptin signaling [74]. Furthermore, the consumption of HFD triggers hypothalamic neuroinflammation by activating the Toll-like receptor 4 (TLR4) signaling pathways [86,87]. In addition to the increased levels of proinflammatory cytokines, several inflammatory and cellular stress responses, including endoplasmic reticulum (ER) stress, SOCS3, and the IKKβ/NF-κB pathways, have been shown to be upregulated [79]. Furthermore, HFD-induced obesity leads to microglia and astrocyte activation and contributes greatly to the neuroinflammatory tone [88,89,90,91].

Recent studies have also demonstrated that HFD consumption induces hypothalamic neuroinflammation, even before peripheral inflammation, insulin resistance, and obesity [64,73]. Hypothalamic inflammation was evident in rodent models of HFD within 1 to 3 days of HFD onset, before substantial weight gain, peripheral inflammation, and peripheral insulin resistance [73]. Furthermore, inhibiting microglia activation and blocking neuroinflammatory pathways in the hypothalamus prevents diet-induced obesity and the metabolic consequences associated with obesity [64,92,93,94].

#### 3.1.2. Hippocampal Neuroinflammation

The hippocampus is a brain region primarily associated with learning and memory mechanisms. The hippocampus, located in the inner area of the temporal lobe, forms part of the limbic system, which is particularly important in regulating emotional responses. Hippocampal impairments can be found in the early phases of neurodegenerative dementias, including vascular dementia and Alzheimer’s disease. The hippocampus is very susceptible to damage by dietary factors, obesity, and metabolic disease [9,57,59,60,61,95]. Dietary fats are among the most proinflammatory components of the obesogenic diet. Saturated fatty acids induce the negative consequences of obesity, while monounsaturated fatty acids promote metabolic health.

Increased hippocampal neuroinflammation, induced by obesity or high-caloric diets, is correlated with deficits in learning and memory [95] and exacerbates cognitive decline in animal models of Alzheimer’s disease [96,97]. The cerebral cortex, also called gray matter, is the brain’s outermost layer, which plays a key role in many high-level functions, such as reasoning, emotion, problem solving, memory, language, and consciousness. Some studies have also demonstrated that HFD-induced obesity activates neuroinflammation mechanisms with harmful effects on the cerebral cortex [98,99,100].

In a genetic model of Alzheimer’s disease (APOE3 knock-in mice), the consumption of an HFD (40% fat) increased hippocampal gliosis as measured using GFAP and Iba1 immunostaining, which supports the hypothesis that the early dysregulation of inflammation could predispose to brain damage [97]. In the 3xTg-AD mice model, an HFD (60% fat) consumption elicited more severe astrogliosis, metabolic dysfunction, and weight gain than in wild-type mice [96]. Mice fed on an HFD for three months had increased hippocampal inflammatory cytokine production and a loss of synaptic protein expression. Dietary obesity impaired hippocampus-dependent memory, reduced long-term potentiation (LTP), and induced the expression of the activation marker major histocompatibility complex II (MHCII) in hippocampal microglia [58]. Rats fed an HFD had memory impairments, an effect that is augmented with a longer duration of HFD consumption and that is linked to elevated levels of IL-1β in the hippocampus [95].

In a rat model of a cafeteria diet supplemented with sucrose, animals exhibited increased hippocampal inflammation, with increased mRNA levels of TNF-α and IL-1β. In this model, the authors observed a strong negative correlation between TNF-α expression and memory performance [68]. Obesogenic western diets induced the expression of hippocampal cytokine levels (IL-1 β, TNF-α, and IL-6) and significantly impacted microglia morphology (increased microglia-shaped activation) [101]. High-fat diet consumption rapidly triggers hippocampal dysfunction associated with neuroinflammation, promoting a progressive breakdown of synaptic and metabolic functions. The consumption of a HFD for only three days induced cognitive deficits, increasing the hippocampus levels of TNF-α and IL-6. Depressive-like behavior was observed after day 5 of HFD consumption. Changes in proinflammatory cytokines accompanied changes in blood–brain barrier permeability [57]. This study highlights that hippocampus inflammation results from local cytokine signals in response to HFD, similar to hypothalamic inflammation [57,65,73].

In a genetic model of obesity, leptin-receptor-deficient *db/db* mice exhibited increased microglia activation (Iba1 measurements) with MHCII immunoreactivity and increased levels of IL-1β in the hippocampus. The levels of IL-1β presented a correlation between adiposity and cognitive impairment [102]. Inhibiting neuroinflammation using the intrahippocampal IL-1 receptor antibody or exercise prevented hippocampal microgliosis, synaptic dysfunction, and cognitive impairment [102,103]. Similar results were found in a rat model of high-fat-diet-induced obesity, in which treadmill exercise decreased the production of proinflammatory cytokines (IL-1 β and TNF-α) and cyclooxygenase-2 (COX-2), as well as inhibiting the TLR4 pathway (myeloid differentiation 88 and tumor necrosis factor receptor-associated factor 6 and the phosphorylation of transforming growth factor β-activated kinase 1, IkBα, and NF-Κb) [103].

In conclusion, neuroinflammation in the hippocampus is a potential mechanism for cognitive deficits induced by obesity and metabolic disorders.

### 3.2. Obesity-Induced Cognitive Decline: Role of Mitochondrial Dysfunction

Diabetes and obesity are modifiable risk factors for cognitive dysfunction and dementia. Several studies have demonstrated and identified overlapping neurodegenerative mechanisms observed in these disorders, including oxidative stress, mitochondrial dysfunction, and inflammation. An excessive intake of nutrients provokes the mitochondria to become overloaded with fatty acids and glucose, leading to an increase in the production of acetyl-CoA. This then causes the production of NADH through the Krebs cycle, which promotes an increase in the electron transfer chain in the mitochondria and, subsequently, increases ROS production, leading to oxidative stress. In addition, there is evidence that the brain’s energy status is decreased in obesity, although the underlying mechanisms are currently unknown. Patients with obesity have been found to have impaired cerebral energy gain upon experimentally increased blood glucose levels up to a postprandial status. This suggests that the brains of individuals with obesity cannot generate an appropriate amount of energy due to dysfunctional glucose transport or a downregulated energy synthesis in mitochondrial respiration [104].

More recent evidence highlights dietary fat’s impact on brain function and cognitive deficits, demonstrating that the long-term consumption of HFD induces impairments in mitochondrial brain function and brain insulin resistance [6,105,106,107,108]. Mitochondrial dysfunction is related to inflammation and other energy-dependent disturbances, where the generation of ROS exceeds the physiological antioxidant protective activity, causing cellular oxidative damage. Different animal models of obesity and metabolic dysfunction have demonstrated impairments in mitochondrial dysfunction and increased oxidative stress.

In a mouse model of diet-induced obesity and insulin resistance, in the brains of 16-week-old mice, the opening of the mitochondrial permeability transition pore (mPTP), the loss of mitochondrial membrane potential (ΔΨm), and apoptosis were observed, while insulin addition ameliorated these dysfunctions. An increase in fission-related proteins and the activation of mitophagy were also detected, which indicate that an alteration in the insulin pathway affects mitochondrial integrity and effective mitophagy [109]. In a type 1 diabetes rat model (streptozotocin, STZ), it was observed that oxygen consumption in basal state 4 significantly increased in the mitochondria from the hippocampus, cortex, and cerebellum. No relevant differences were observed in the activity of respiratory complexes, but hippocampal mitochondrial membrane potential was reduced [110]. Moreover, ADP-stimulated state 3 respiration and uncoupled maximal respiration (FCCP) were significantly decreased in the hippocampus of diabetic rats compared to controls [111]. Sirtuin-1 (SIRT1) is a major regulator of mitochondrial biogenesis and metabolism through the activation of proliferator-activated receptor gamma coactivator-1 α (PGC-1α). SIRT-1 and PGC-1α were also significantly decreased in the hippocampus of diabetic rats, which suggests an impairment in mitochondrial biogenesis and, consequently, a reduction in the mitochondrial respiratory capacity [111]. Sirtuin-3 (SIRT3) plays a significant role in enhancing mitochondrial protein function. The downregulation of SIRT3 is a key component of metabolic syndrome, a precondition for obesity and diabetes. In a genetic model of metabolic syndrome, Sirt3−/− mice fed a western diet had impaired brain mitochondrial respiration, lower levels of mitochondrial fission proteins Mfn1 and Mfn2, and hyperacetylated brain mitochondrial proteins [112]. The brains of Sirt3−/− mice also presented the downregulation of enzymes in several metabolic pathways, including fatty acid oxidation and the tricarboxylic acid cycle [112].

It was demonstrated that Zucker diabetic fatty (ZDF, FA/FA) rats had an increased ROS production in the brain, as well as increased nitric oxide (NO) production [113]. In addition, catalase activity in the brains of ZDF rats was significantly reduced. Compared with the lean model, glutathione metabolism and mitochondrial respiratory functions were negatively affected in ZDF rats. Specifically, complex I, II/III, and IV activities and mitochondrial ATP content were significantly decreased, indicating that the brain developed complications associated with redox homeostasis and mitochondrial dysfunction [113].

Rats fed an HFD for 12 weeks had increased brain and hippocampus ROS production (malondialdehyde, dichlorohydrofluoresceindiacetate (DCFHDA), and H_2_O_2_ levels), lipid peroxidation, brain mitochondrial depolarization (JC-1 marker), and brain mitochondrial swelling that results in cognitive decline [108,114,115,116]. It was observed that energy restriction combined with a dipeptidyl peptidase-4 inhibitor (vildagliptin) or vildagliptin alone for four weeks restored brain mitochondrial function, hippocampal synaptic plasticity, and cognitive function [115,116,117]. Vagal nerve stimulation therapy in rats fed an HFD for 12 weeks promoted an improvement in brain insulin sensitivity, decreased ROS production, attenuated mitochondrial brain dysfunction and cell apoptosis, and consequently improved cognitive function [118]. 

A mice model of HFD-induced obesity (50% fat for 18 weeks) demonstrated changes in brain cortex bioenergetics, with changes in mitochondrial function, efficiency, and oxidative stress [98]. Specifically, the authors showed that the brains of HFD animals used fatty acid as a preferential fuel source compared to control animals. Since dysfunctional synaptic mitochondria may lead to impaired neurotransmission and cognitive failure, using synaptosomal fractions demonstrated a decrease in basal respiration and the maximal rate of respiration and ATP production in HFD mice. The same study also revealed a significant reduction in the proton leak in the HFD mice [98]. In these animals’ cortex, glutathione (GSH) content significantly decreased compared to that in controls, and the HFD mice exhibited a lower GSH/GSSG ratio than the controls [98]. High-fat diet-induced obesity increased oxidative stress in the hippocampus with increased mitochondrial peroxide (H_2_O_2_) production. Mitochondria respiration (oxygen consumption) was significantly lower in obese mice than in lean mice [119]. Treadmill running for 12 weeks improved short-term memory and oxidative stress (decreased mitochondrial H_2_O_2_ production). The exercised animals had a higher mitochondrial O_2_ respiration capacity than that of a sedentary model of diet-induced obesity. Mitochondrial Ca^2+^ retention capacity was higher in the mice that performed exercise and, consequently, decreased mitochondrial permeability transition pore opening sensitivity and neuronal apoptosis [119].

The consumption of HFDs for only two weeks did not affect the mitochondrial activity in the hippocampus (analyzed using high-resolution respirometry). However, four weeks of HFD feeding induced mitochondrial dysfunction in hippocampal homogenates [57]. Four weeks of the consumption of HFD induced a significantly lower O_2_ consumption and caused a significant reduction in the electron transfer chain [57]. These animals had depressive-like behavior and changes in memory function after five days of HFD consumption [57].

In mice fed obesogenic diets (high-fat and high-glycemic diets) for 12 weeks, it was observed that changes in the brain cytoskeletal proteins, mitochondria, and metabolic proteins changed their post-translational status [120]. Specifically, proteins involved in mitochondrial functions were downregulated in mice fed obesogenic diets compared to lean mice. This suggests a reduced metabolism and a lower activity of mitochondria in obese mice [120]. In the brains of obese mice, lower expressions in nine proteins involved in mitochondrial activity were observed, namely, dehydrogenase [ubiquinone] iron-sulfur proteins 4 and 5 (NDUFS4 and NDUFS5, respectively) and cytochrome c oxidase subunit 7B, mitochondrial (COX7) (components of complex I and complex IV of the respiratory chain) [120]. Alterations in the mitochondrial respiration chain cause impairments in the functionality of the cells. The glutaredoxin (GRX) enzymes are glutathione-dependent oxidoreductases with important roles in regulating cytosolic thiol/disulfide balance and protecting proteins from oxidative damage. The hippocampus of GRX2-knockout mice fed a high-fat diet had worse oxidative stress and mitochondrial impairment than wild-type mice. These results provide evidence that preventing oxidative damage may have protective functions in HFD-elicited brain injury and cognitive deficits [121].

In the hypothalamus, mitochondrial dynamics regulate energy homeostasis and metabolism [122,123]. Although inflammation is one of the mechanisms involved in hypothalamic neuronal defects in diet-induced obesity, mitochondrial abnormalities also occur [65,124,125,126]. In the hypothalamus, mitochondrial fusion regulates neuronal firing via the modulation of intracellular ATP levels in diet-induced obesity. The deletion of Mfn2 in the anorectic pro-opiomelanocortin (POMC) neurons of the hypothalamus disrupts endoplasmic reticulum (ER)–mitochondria contacts, ER stress activation, leptin resistance, and obesity (Schneeberger et al., 2013). However, the deletion of Mfn2 in orexigenic agouti-related peptide (Agrp) neurons induced less weight gain in mice fed a HFD [126]. High-fat diet consumption for one day caused a transient reduction in Mfn2, although after seven days of the consumption of this diet, the expression of Mfn2 increased [65]. HFD diet consumption for five weeks promoted the upregulation of Mfn2 in the arcuate nucleus and downregulation after 13 weeks of HFD consumption [127]. Using an in vitro model of hypothalamic neuronal cells, the authors demonstrated that saturated palmitic acid decreased the protein levels of mitofusin-2 and activated the ER stress response, exacerbating insulin resistance in hypothalamic neuronal cells [127]. Neuronal cell treatment with a high concentration of palmitic acid increased mitochondria ROS production. Insulin resistance induced by palmitic acid was prevented by treatment with the anti-inflammatory reagent and ER stress inhibitors [127].

Under physiological conditions, the redox signaling pathway initiates the hypothalamus glucose-sensing mechanism. Obese Zücker rats had cerebral hypersensitivity to glucose, leading to abnormal vagus-induced insulin secretion [128]. The impaired hypothalamic glucose sensing in obese Zücker rats is linked to abnormal redox signaling, which originates from mitochondria dysfunction. In the hypothalamus of obese Zücker rats, abnormal glutathione redox state and increased ROS production occur. In this animal model, mitochondria activity in complexes I and IV in the hypothalamus was significantly increased. The total respiratory capacity was also increased. The expressions of complexes I, II, III, and IV were raised in the hypothalamic mitochondria of obese rats, and no changes were observed in complex V [128].

In a comparison of a diet-induced obesity-resistant mouse strain (WSB/EiJ to the obesity-prone mouse strain (C57BL/6J strain), it was possible to observe that the resistant strain displayed a lower inflammatory status, both peripherally and centrally (less activated microglia in the hypothalamus), and more reactive and responsive mitochondria in the hypothalamus. In the hypothalamus of the resistant strain, an increase occurred in ATP content after eight weeks of a high-fat diet [129]. This indicates that the reduced inflammatory responses and increased mitochondrial activity contribute to diet-induced obesity resistance. Independent of the mouse strain, eight weeks of high-fat diet feeding induces a decrease in each mitochondria area [122].

The consumption of a western diet increases the circulating levels of palmitate, which is converted into ceramide in order to accumulate in tissues in response to obesity. Pharmacological and genetic strategies that reduce tissue ceramide levels reverse the metabolic consequences of obesity. Specifically, reducing ceramide production protects mice from the metabolic effects of high-fat diet consumption by preventing the fragmentation of the mitochondrial network within the hypothalamus. Restoring mitochondrial function through the decreased accumulation of ceramide increases leptin sensitivity and, consequently, reduces food intake [130].

Targeting mitochondrial dysfunction and oxidative stress could have potential benefits for cognitive dysfunction. N-acetylcysteine (NAC) is used to treat several diseases related to oxidative stress and inflammation due to its antioxidant and anti-inflammatory properties [131]. In a rat model of high-fat diet-induced obesity, NAC treatments improve glycemia and peripheral insulin resistance and reduce the oxidative stress/neuroinflammation/inflammasome activation axis in the cerebral cortex [132]. In a mouse model of obesity, NAC prevents hippocampal alterations and memory impairment [133]. In older humans, GlyNAC (a combination of glycine and N-acetylcysteine NAC) supplementation for 24 weeks decreases oxidative stress and mitochondrial dysfunction; reduces inflammation, insulin resistance, and endothelial dysfunction; and improves cognition [134]. Although additional studies are needed to address these effects in humans, these studies suggest that treatments, such as NAC, that reduce oxidative stress and improve mitochondria could be used as a simple and viable method to promote brain health.

In summary, various studies have demonstrated that diet-induced obesity and metabolic disorders induce mitochondrial dysfunction and oxidative stress in the brain, contributing to neuronal dysfunction, the dysregulation of whole-body metabolism, and cognitive deficits.

## 4. Conclusions

In summary, the findings suggest that high-fat and high-caloric diet consumption, as well as obesity and associated type 2 diabetes, triggers brain dysfunction associated with neuroinflammation and mitochondrial dysfunction. Neuroinflammation and mitochondrial dysfunction could be some of the mechanisms involved in cognitive deficits and dementia associated with obesity and metabolic disorders. Treatments that ameliorate brain mitochondrial dysfunction and decrease hippocampal oxidative stress levels have beneficial effects on cognitive processes. In addition to elucidating the link between diet and cognitive function, it might be relevant to comprehend the neurodegenerative process.

## Figures and Tables

**Figure 1 metabolites-13-00086-f001:**
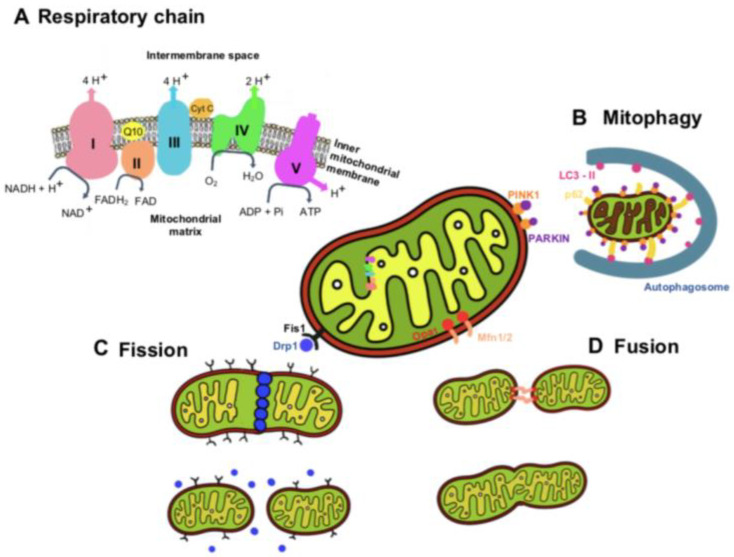
Respiratory chain and mitochondrial dynamics. (**A**) Respiratory chain. Electrons and protons flow through complexes of the respiratory chain in oxidative phosphorylation. (**B**) Mitophagy is mediated by PINK1, PARKIN, and P-62 proteins, which recruit the protein LC3-II and form the autophagosome for cell degradation. (**C**) Mitochondrial fission is mediated by the protein Drp1, which is recruited from the cytosol to interact with the protein fis1 in the mitochondrial outer membrane, forming constriction points that lead to mitochondrial fission. (**D**) Mitochondrial fusion requires the action of the Opa1 protein on the inner mitochondrial membrane and the action of Mfn1 and Mfn2 proteins in the outer mitochondrial membrane, promoting the fusion of juxtaposed mitochondrial membranes.

**Figure 2 metabolites-13-00086-f002:**
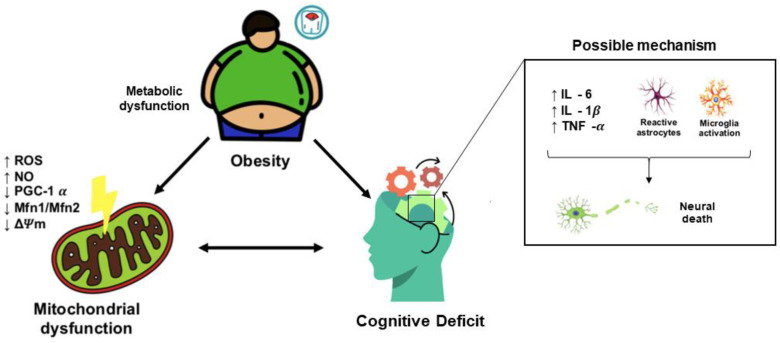
Obesity, neuroinflammation, and mitochondrial dysfunction. Excessive food consumption in obesity can lead to mitochondrial dysfunction characterized by increased reactive oxygen species (ROS) levels, increased nitric oxide (NO) levels, decreased protein content of PGC-1α and Mfn1/Mfn2, and decreased mitochondrial membrane potential (ΔΨm). Obesity is associated with increased levels of inflammatory cytokines in the brain and compromises neural viability.

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
