# Peer review of "Obesity-Induced Brain Neuroinflammatory and Mitochondrial Changes"

_metabolites, 2023, doi:10.3390/metabo13010086_

Round 1

Reviewer 1 Report

This review article provides an interesting view on the interaction between metabolic homeostasis and neuroimmunological regulation in the development of obesity associated brain dysfunction. I suggest a few minor revisions, and propose points that could be considered for better understanding of the authors' view and interpretation of the literature.

Page 2, lines 47-49: The authors note that obesity increases AD risk. However, overall dementia risk and vascular dementia have also been associated to obesity and obesity-related comorbidities and prediabetes.

The authors covered pathological aspects of mitochondria dysfunction in inflammation and obesity. However, given the topic, it would be interesting to note how mitochondrial functions and dynamics are modulated by cytokine and hormone (insulin, leptin) signaling in health.

In the realm of mitochondrial dysfunction, the authors should provide a brief note that there's evidence provided by treatments with the widely used antioxidant N-acetylcysteine in models of obesity and diabetes. A recent review on the topic: https://pubmed.ncbi.nlm.nih.gov/32390631/

A few examples of recent evidence:

NAC ameliorating AD pathology (https://www.mdpi.com/2076-3921/10/6/967) and HFD deterimental effects (https://pubmed.ncbi.nlm.nih.gov/31838118/), as well as HFD-induced memory impairment (https://pubmed.ncbi.nlm.nih.gov/36222315/).

Inflammation in diet-induced obesity: can the authors clarify whether inflammation in the CNS results from peripheral cytokines, or from local cytokine signals in response to other type of obesity cues? or maybe their combination?

Lines 234-250 and 324-325: what are the molecular mediators of anti-inflammatory actions of exercise?

The authors conclusion should keep focus on what's covered in the manuscript. In particular, in my view, the review does not effectively address how neuroinflammation and mitochondrial dysfunction specifically promote synaptic dysfunction. The authors wrote: "high-fat and high-caloric diet consumption, as well as obesity, triggers hippocampal dysfunction associated with neuroinflammation, and mitochondrial dysfunction, promoting a progressive breakdown of synaptic function."

Minor:'

line 312: 'mice' should be 'mouse'

Author Response

We are grateful for the reviewers' positive, constructive, and helpful comments. The reviewers pointed out some issues that were not clearly explained in the previous version of our manuscript, and new topics were added to improve the manuscript. As a result, the manuscript is now corrected in the revised version. Hopefully, we addressed all the reviewers' concerns.

Reviewer #1

This review article provides an interesting view of the interaction between metabolic homeostasis and neuroimmunological regulation in the development of obesity associated brain dysfunction. I suggest a few minor revisions, and propose points that could be considered for better understanding of the authors' view and interpretation of the literature.

Point 1- Page 2, lines 47-49: The authors note that obesity increases AD risk. However, overall dementia risk and vascular dementia have also been associated to obesity and obesity-related comorbidities and prediabetes.

Response to point 1:  We agreed with the reviewer's comment and added a more detailed discussion in the revised manuscript.

Line 42: "In recent years, increasing attention has been given to the relationship between obesity and associated insulin resistance/type 2 diabetes with the development of brain diseases, including depression, neurodegenerative diseases, dementia, and vascular dementia [8-10]. Epidemiological studies showed that people with higher BMI are at greater risk for developing Alzheimer's disease than subjects with normal BMI [11-13]. Also, some population-based studies have identified diabetes as a risk factor for dementia and metabolic syndrome, a grouping of risk factors for type 2 diabetes mellitus [14-16]. However, cellular and molecular mechanisms linking these conditions are not yet fully elucidated."

Point 2- The authors covered pathological aspects of mitochondria dysfunction in inflammation and obesity. However, given the topic, it would be interesting to note how mitochondrial functions and dynamics are modulated by cytokine and hormone (insulin, leptin) signaling in health.

Response to point 2: Thanks for the interesting comment. However, very few studies have analyzed the exact role/ mechanism of insulin or leptin in the brain's mitochondrial dynamics. Moreover, most studies used HFD-induced obesity, which is not a clean model to analyze the single effect of each hormone in the mitochondrial dynamics. Nevertheless, both hormones seem to regulate the process, and we added some information in the revised manuscript.

Line 104: ". Mitochondrial dynamics involve the plasma membrane and organelles, such as ER and lysosomes. The contact point of ER-mitochondria is referred to as mitochon-dria-associated ER membranes. Some studies have suggested that the integrity of mi-tochondria-associated ER membranes is required for insulin signaling(for a detailed description, see the revision [24]). Effect/defect of insulin signaling in different features of mitochondrial dysfunction, focusing on dynamics, biogenesis, and mitophagy and their role in pathologies in which metabolic dysmetabolism is comorbidity with neurodegeneration [25, 26]. Some studies also suggest that the protective actions of leptin may be facilitated through the regulation of mitochondrial dynamics, namely mitochondrial fission and fusion [27-30]."

Line 338: " In the brain of 16 weeks of age mouse model of diet-induced obesity and insulin re-sistance was observed the opening of mitochondrial permeability transition pore (mPTP), loss of mitochondrial membrane potential (ΔΨm) loss, and apoptosis while insulin addition ameliorated these dysfunctions. Was also detected an increase of fission-related proteins and activation of mitophagy, which indicate that alteration of the insulin pathway affects mitochondrial integrity, and effective mitophagy [109]. "

Point 3- In the realm of mitochondrial dysfunction, the authors should provide a brief note that there's evidence provided by treatments with the widely used antioxidant N-acetylcysteine in models of obesity and diabetes. A recent review on the topic: https://pubmed.ncbi.nlm.nih.gov/32390631/

A few examples of recent evidence:

NAC ameliorating AD pathology (https://www.mdpi.com/2076-3921/10/6/967) and HFD deterimental effects (https://pubmed.ncbi.nlm.nih.gov/31838118/), as well as HFD-induced memory impairment (https://pubmed.ncbi.nlm.nih.gov/36222315/).

Response to point 3:  We agreed with the reviewer's comment and added a more detailed discussion in the revised manuscript.

Line 468: " Targeting mitochondrial dysfunction and oxidative stress could have potential benefits for cognitive dysfunction. N-acetylcysteine (NAC) is used to treat several diseases related to oxidative stress and inflammation due to its antioxidant and an-ti-inflammatory properties [131]. In a rat model of high-fat diet-induced obesity, NAC treatments improve glycemia and peripheral insulin resistance and reduce oxidative stress/neuroinflammation/inflammasome activation axis in the cerebral cortex[132]. In a mouse model of obesity, NAC prevents hippocampal alterations and memory impairment [133]. In older humans, GlyNAC (combination of glycine and N-acetylcysteine NAC) supplementation for 24 weeks decreases oxidative stress and mitochondrial dysfunction, reduces inflammation, insulin resistance, and endothelial dysfunction, and improves cognition [134]. Although additional studies are needed to address these effects in humans, these suggest that treatments, such as NAC, that reduce oxidative stress and improves mitochondria could be used as a simple and viable method to promote brain health. "

Point 4- Inflammation in diet-induced obesity: can the authors clarify whether inflammation in the CNS results from peripheral cytokines, or from local cytokine signals in response to other type of obesity cues? or maybe their combination?

Response:  We agreed with the reviewer's comment and added a more detailed discussion in the revised manuscript.

Line 246: " In addition to the increased levels of proinflammatory cytokines, several inflammatory and cellular stress responses, including endoplasmic reticulum (ER) stress, SOCS3, and the IKKβ/NF-κB pathways, were shown to be upregulated [79]. Furthermore, HFD di-et-induced obesity leads to microglia and astrocyte activation and contributes greatly to the neuroinflammatory tone [88-91].

            Recent studies have also demonstrated that HFD consumption induces hypothalamic neuroinflammation even before peripheral inflammation, insulin resistance, and obesity [64, 73]. Hypothalamic inflammation was evident in rodent models of HFD within 1 to 3 days of HFD onset, before the substantial weight gain, peripheral inflammation, and peripheral insulin resistance [73]."

Line 292: " High-fat diet consumption rapidly triggers hippocampal dysfunction associated with neuroinflammation, promoting a progressive breakdown of synaptic and metabolic functions. Consumption of a high-fat diet for only three days induces cognitive deficit, increasing the hippocampus levels of TNF-α and IL-6. Depressive-like behavior was observed after day 5 of HFD consumption. Changes in proinflammatory cytokines accompanied changes in blood-brain barrier permeability[57]. This study highlights that hippocampus inflammation results from local cytokine signals in response to high-fat diets, similar to hypothalamic inflammation [57, 65, 73]."

Point 5- Lines 234-250 and 324-325: what are the molecular mediators of anti-inflammatory actions of exercise?

Response to point 5: Thanks for the comment. We added the molecular mediator in the new version of the manuscript.

Line 391: " Mitochondria respiration (oxygen consumption) was significantly lower in obese mice than in lean mice [119]. Treadmill running for 12 weeks improved short-term memory and oxidative stress (decreased mitochondrial H2O2 production). Exercised animals had higher mitochondrial O2 respiration capacity compared to a sedentary model of diet-induced obesity. Mitochondrial Ca2+ retention capacity was higher in the mice that performed exercise and consequently had decreased mitochondrial permeability transition pore opening sensitivity and decreased neuronal apoptosis [119]."

Line 306: " Similar results were found in a rat model of high-fat diet-induced obesity, in which treadmill exercise decreased the production of proinflammatory cytokines (IL-1 β, TNF-α), cyclooxygenase-2 (COX-2), and also by inhibiting the TLR4 pathway (myeloid differentiation 88 and tumor necrosis factor receptor-associated factor 6, and phosphorylation of transforming growth factor β-activated kinase 1, IkBα and NF-Κb) [103]."

Point 6- The authors conclusion should keep focus on what's covered in the manuscript. In particular, in my view, the review does not effectively address how neuroinflammation and mitochondrial dysfunction specifically promote synaptic dysfunction. The authors wrote: "high-fat and high-caloric diet consumption, as well as obesity, triggers hippocampal dysfunction associated with neuroinflammation, and mitochondrial dysfunction, promoting a progressive breakdown of synaptic function."

Response to point 6:  We agreed with the reviewer's comment and changed the conclusion.

Line 485: " In summary, findings suggest that high-fat and high-caloric diet consumption, as well as obesity and associated type 2 diabetes, triggers brain dysfunction associated with neuroinflammation and mitochondrial dysfunction. Neuroinflammation and mitochondrial dysfunction could be some of the mechanisms involved in cognitive deficits and dementia associated with obesity and metabolic disorders."

Minor:'

line 312: 'mice' should be 'mouse'

These were corrected in the manuscript.

Reviewer #2

This manuscript is a review addressing the effects of obesity on brain neuroinflammation and mitochondrial changes as a link between metabolic dysfunction and cognitive impairments. I have a few comments that I would like the authors to address before I can recommend the acceptance of this manuscript.

Major Concerns:

Point 1. The authors only briefly mentioned that obesity is associated with increased levels of inflammatory cytokines and immune cells in the brain. However, I think it is very desirable for the authors to provide some more details about the related molecular and cellular mechanisms underlying obesity-induced brain neuroinflammation. 

Response point 1:  We agreed with the reviewer's comment and added a more detailed discussion in the revised manuscript.

Line 200: "Systemic low-grade chronic inflammation has been reported to cause neuroinflamma-tion and changes in different brain structures, such as the cerebellum, amygdala, cerebral cortex, and hypothalamus [57, 73-75]. Obesity-induced inflammation has been related to changes in the integrity of the blood-brain barrier permeability, inducing leukocyte extravasation along with the potential entry of pathogens and toxins into the central nervous system, which in turn stimulate more inflammatory responses in a vicious cycle [57]. The decrease in tight junction protein expression and disturbed blood-brain barrier are regulated by the NF-κB pathway increasing the expression of proinflammatory proteins such as IL-1β, TNFα, and IL-6. Loss of the blood-brain barrier during obesity facilitates proinflammatory cytokines to enter the brain parenchyma, thus allowing them to interact and activate glial cells (microglia and astrocytes)[76]. Activated microglia secrete more inflammatory cytokines (TNFα, IL-1β, IL-6), perpetuating the neuroinflammation and leading to neuronal damage. NLRP3 proteins of the inflammasome secreted by the visceral adipose tissue directly activate microglia through the IL1 receptor [77].

            Furthermore, high-fat diets can directly activate microglial cells, inducing morphological changes in the hypothalamus without microglial changes in the cerebral cortex and striatum. Also, HFD diet induced-obesity is associated with increased peripheral immune cells entry into the central nervous system and may contribute to the inflammatory response [78]. Fatty acid intake induces the activation of immune cells and inflammatory response through the activation of the innate immune system through Toll-like receptors (TLRs)[63]. The binding of fatty acids to TLR4 activates nuclear factor-Κb (NF-Κb) and activator protein 1 (AP-1) that, in turn, upregulate the expression of proinflammatory cytokines and chemokines [79]. Another proposed mechanism for obesity-induced inflammation relies on the ability of an HFD to modulate the gut microbiota. Indeed, the subsequent changes in microbiota populations result in the permeabilization of the gut barrier leading to the increased passage of bacterial endotoxins into the circulation [80, 81]."

Point 2. Figure 2 is too simple and doesn't mention cognitive impairment. The author should revise Figure 2 or show an additional figure to clearly describe how diet-induced obesity or metabolic dysfunction leads to cognitive decline via brain neuroinflammation and mitochondrial changes. This will contribute significantly to the main goal of this review.

Response to point 2: Thanks for the suggestion. We added a modified figure 2.

Point 3. There are many grammatical errors throughout the article that make it difficult to read. Additionally, there are a few spelling errors that need to be addressed.  For example, Line 32: "24,9" should be "24.9". Line 230: "Mice fat on a HFD".

Response Point 3: Thanks for the comment. We revised the entire manuscript.

Minor Comments:
Point 4: Line 36-37. References 1 and 2 were cited to support that "More alarming is that, in 2020, around 39 million children under the age of 5 were overweight or obese". However, these two papers were published in 2019 and 2017, respectively. This is misleading and doesn't make sense.

Response Point 4: These were corrected in the manuscript. " In 2015, more than 1.9 billion adults were overweight; over 600 million were obese [1, 2]. More alarming is that child obesity affects 107.7 million children [1, 2]. "

Point 5. Line 135-138."An 18-year follow-up longitudinal study demonstrated that women who developed dementia between 79 and 88 years old were overweight. Also, women with a higher BMI developed dementia earlier, around 70 years old, than non-demented women [32]." These two sentences should be rewritten.

Response point 5: The sentence was corrected in the manuscript. " An 18-year follow-up longitudinal study demonstrated a higher degree of overweight in older women who developed AD. No associations were not found in men [42]. "

Point 6. Line 234-236. "Rat fed a HFD had impaired memory, effect that was augmented with a longer duration of HFD consumption, and is linked to elevated levels of IL-1 in the hippocampus". Grammar errors should be corrected.

Response point 6: The sentence was corrected in the manuscript. " Rats fed an HFD had memory impairments, an effect augmented with a longer duration of HFD consumption and is linked to elevated levels of IL-1β in the hippocampus [95]."

Reviewer 2 Report

This manuscript is a review addressing the effects of obesity on brain neuroinflammation and mitochondrial changes as a link between metabolic dysfunction and cognitive impairments. I have a few comments that I would like the authors to address before I can recommend the acceptance of this manuscript.

Major Concerns:

1. The authors only briefly mentioned that obesity is associated with increased levels of inflammatory cytokines and immune cells in the brain. However, I think it is very desirable for the authors to provide some more details about the related molecular and cellular mechanisms underlying obesity-induced brain neuroinflammation. 

2. Figure 2 is too simple and doesn’t mention cognitive impairment. The author should revise Figure 2 or show an additional figure to clearly describe how diet-induced obesity or metabolic dysfunction leads to cognitive decline via brain neuroinflammation and mitochondrial changes. This will contribute significantly to the main goal of this review.

3. There are many grammatical errors throughout the article that make it difficult to read. Additionally, there are a few spelling errors that need to be addressed.  For example, Line 32: “24,9” should be “24.9”. Line 230: “Mice fat on a HFD”.

Minor Comments:

1. Line 36-37. References 1 and 2 were cited to support that “More alarming is that, in 2020, around 39 million children under the age of 5 were overweight or obese”. However, these two papers were published in 2019 and 2017, respectively. This is misleading and doesn’t make sense.

2. Line 135-138. “An 18-year follow-up longitudinal study demonstrated that women who developed dementia between 79 and 88 years old were overweight. Also, women with a higher BMI developed dementia earlier, around 70 years old, than non-demented women [32].” These two sentences should be rewritten.

3. Line 234-236. “Rat fed a HFD had impaired memory, effect that was augmented with a longer duration of HFD consumption, and is linked to elevated levels of IL-1 in the hippocampus”. Grammar errors should be corrected.

Author Response

(The authors gave the same response as above.)
